# Combination of Sanguisorbigenin and Conventional Antibiotic Therapy for Methicillin-Resistant *Staphylococcus aureus*: Inhibition of Biofilm Formation and Alteration of Cell Membrane Permeability

**DOI:** 10.3390/ijms23084232

**Published:** 2022-04-11

**Authors:** Shu Wang, Xiang-Qian Liu, Ok-Hwa Kang, Dong-Yeul Kwon

**Affiliations:** 1Department of Oriental Pharmacy, College of Pharmacy and Wonkwang Oriental Medicines Research Institute, Wonkwang University, Iksan 54538, Korea; wshu1996@gmail.com (S.W.); kangokhwa@daum.net (O.-H.K.); 2School of Pharmacy, Hunan University of Chinese Medicine, Changsha 410208, China; lxq0001cn@163.com

**Keywords:** MRSA, synergy, biofilm, membrane, *Sanguisorba officinalis* L.

## Abstract

Methicillin-resistant *Staphylococcus aureus* (MRSA) infection is challenging to eradicate because of antibiotic resistance and biofilm formation. Novel antimicrobial agents and alternative therapies are urgently needed. This study aimed to evaluate the synergy of sanguisorbigenin (SGB) isolated from *Sanguisorba officinalis* L. with six conventional antibiotics to achieve broad-spectrum antibacterial action and prevent the development of resistance. A checkerboard dilution test and time-to-kill curve assay were used to determine the synergistic effect of SGB combined with antibiotics against MRSA. SGB showed significant synergy with antibiotics and reduced the minimum inhibitory concentration of antibiotics by 2–16-fold. Biofilm inhibition assay, quantitative RT-PCR, crystal violet absorption, and transmission electron microscopy were performed to evaluate the synergy mechanism. The results indicated that SGB could inhibit biofilm formation and alter cell membrane permeability in MRSA. In addition, SGB was found to exhibit quite low cytotoxicity and hemolysis. The discovery of the superiority of SGB suggests that SGB may be an antibiotic adjuvant for use in combination therapy and as a plant-derived antibacterial agent targeting biofilms.

## 1. Introduction

Methicillin-resistant *Staphylococcus aureus* (MRSA) is a multidrug-resistant bacterial pathogen that is a leading cause of hospital and community-acquired infections [1]. Colonization with MRSA leads to skin and soft-tissue infections, bone and joint infections, bacteremia, pneumonia, and endocarditis with persistently high morbidity and mortality [2]. Initially acquired in healthcare settings, MRSA has now emerged in the community in populations without risk factors for MRSA acquisition and has subsequently been detected in colonized or infected animals and foods of animal origin; MRSA transmission between countries has also been frequently reported [3]. The global spread of MRSA has transformed simple incision infections into a potential cause of severe infection [4], greatly reducing treatment options and significantly increasing medical costs [5]. The current first-line treatment of MRSA typically involves the administration of high-dose systemic antibiotics, including vancomycin, linezolid, or ceftaroline; however, clinical isolates resistant to these antibiotics have emerged during the last 20 years [6]. Antibiotic resistance (AMR) threatens global health by preventing effective treatment against bacteria, parasites, viruses, and eukaryotic pathogens, and the misuse or overuse of antibiotics will accelerate the selection of drug-resistant variants [7]. Since all organisms have evolved genetic mutations to prevent lethal selection pressure, AMR is an inevitable evolutionary consequence [8]. Today, resistance to most antibiotics is widespread, and the supply of new drugs has been exhausted [9]. Consequently, alternative strategies and novel antimicrobials are urgently needed to address AMR problems [10].

Biofilms are composed of microorganisms that are adhered to a solid surface and encased in an exopolysaccharide matrix that they have synthesized [11]. Biofilm communities are highly complex and nonreplicating, which defend against multiple clearance mechanisms (e.g., host immune responses), enabling biofilms to survive in harsh environments [12]. In addition, the biofilm matrix, which acts as a bacterial fortress, increases bacterial resistance to antibiotics by reducing the rate of antibiotic diffusion and preventing entry into the biofilm [13]. The rapid development of multidrug-resistant (MDR) bacteria is due to the inappropriate and overuse of antibiotics and the ineffectiveness of antibiotics for difficult-to-treat biofilm-associated infections [14]. The quorum sensing (QS) system is a regulatory system for biofilm formation, and the *hld* gene is one of the most important regulators of the QS system [15]. A large percentage of bacterial infections are considered to be associated with biofilm formation, resulting in a large number of deaths each year [16]. Therefore, antimicrobial agents capable of rapidly penetrating and inhibiting biofilms represent a potentially valuable therapeutic alternative for the treatment of MRSA infections. Moreover, certain antibiotics have minimal antimicrobial activities toward bacteria related to membrane permeability resistance; accordingly, the combination of medications with membrane permeabilizes or cell membrane disruptors is regarded as an effective combination treatment [17].

Plants are renewable and cost-effective sources of antimicrobial agents with low toxicity and little to no drug resistance; hence, they are considered to have clinical value [18]. *Sanguisorba officinalis* L., a member of the subfamily Rosoideae and family Rosaceae, is a perennial plant widely distributed in eastern Asia, North America, and western Europe [19]. In previous work, we demonstrated that triterpenoid saponin sanguisorbigenin (SGB) from the dried root of *Sanguisorba officinalis* L. is a potential plant-derived antimicrobial agent, with an MIC range of 12.5–50 μg/mL against standard strains and clinical isolates [20]. The current study further evaluated the synergy of SGB with six conventional antibiotics including non-β-lactam antibiotics against MRSA. Furthermore, we investigated the inhibition of biofilm formation and the effect of SGB on the permeability of cell membranes.

## 2. Results

### 2.1. Synergistic Interactions of SGB with Conventional Antibiotics

A checkerboard assay was performed to evaluate the double combinations of SGB with six conventional antibiotics against one reference strain ATCC 33591 and two isolates DPS-1 and DPS-3 (Table 1). Among the 18 combinations examined, 11 showed synergy (61.0%), two showed partial synergy (11.0%), five showed additive effects (28.0%), and no antagonism was detected. The FICI values of double combinations of SGB with six conventional antibiotics against MRSA ranged from 0.19 to 1 and reduced the MIC value of antibiotics by 2–16-fold.

### 2.2. Time-to-Kill Assay

The results of a time-to-kill assay of SGB in combination with six conventional antibiotics against *S. aureus* (ATCC 33591 and DPS-1) further corroborated the checkerboard assay results. Compared to the most active single drug group, all combination groups exhibited significant synergistic interactions and more than 3 log_10_ reductions in colony count after 24 h. As shown in Figure 1, none of the antimicrobials alone could completely inhibit bacterial growth after 24 h of incubation. However, except for the combination group with vancomycin (Figure 1c), all the combination treatment groups could ultimately kill the bacteria after 24 h of incubation. Furthermore, SGB combination with linezolid or amoxicillin showed a more significant synergistic interaction than the other four combinations (Figure 1b,c), killing the bacteria completely within 16 h. In addition, except for the gentamicin treatment group (Figure 1a), the antibacterial effect of the subinhibitory concentrations of SGB was more potent than that of conventional antibiotics within 24 h.

### 2.3. SGB Inhibited Biofilm Formation and Downregulated the Expression of the hld

Figure 3a shows the effects of SGB on biofilm formation by clinical MRSA isolates investigated at subinhibitory concentrations (1/8 MIC, 1/4 MIC, and 1/2 MIC). SGB significantly and dose-dependently inhibited biofilm formation at subinhibitory concentrations by more than two-fold. Specifically, at 1/2 MIC, SGB inhibited biofilm formation of ATCC 33591 and DPS-1 by 86% and 91%, respectively (Figure 2a). Additionally, the expression of the biofilm regulatory gene *hld* of MRSA was downregulated significantly after treating with SGB at subinhibitory concentrations (Figure 2b).

### 2.4. SGB Increased Crystal Violet Absorptions

The results of the crystal violet absorption assay are shown in Figure 3. The absorption of crystal violet of *S. aureus* ATCC 33591 treated with 2 MIC and 4 MIC of SGB was increased 2.1-fold and 2.3-fold, respectively (Figure 3a). The absorption of crystal violet of *S. aureus* DPS-1 treated with 2 MIC and 4 MIC of SGB was increased 3.2-fold and 3.5-fold, respectively (Figure 3b). SGB significantly altered the membrane permeability of *S. aureus* and increased crystal violet absorption. As a negative control, vancomycin had no discernible impact, showing that it does not influence membrane permeability.

### 2.5. SGB Damaged the Morphological of MRSA Cells

The results of TEM showed the impact of SGB on the morphology of MRSA ATCC 33591. The untreated MRSA strains were observed to have normal morphology with intact septa (Figure 4a), whereas MRSA strains exposed to 1/2 MIC (6.3 μg/mL) of SGB were observed to have damaged cytoplasmic membranes and rougher surfaces compared to the untreated control MRSA (Figure 4b). Moreover, MRSA exposure to the MIC (12.5 μg/mL) of SGB was shown to lyse the cell and release cytoplasmic contents, with the cells appearing almost absent (Figure 4c).

### 2.6. Cytotoxicity and Hemolysis Activity of SGB

The potential cytotoxicity and hemolysis activity of SGB was investigated to exclude the toxicity of SGB toward mammalian cells. The result showed that the IC_50_ value against RAW 264.7 cells was approximately 100 µg/mL, which is more than 2–8-fold the MIC values of standard strains or clinical strains, respectively (Figure 5a). Moreover, compared to Triton X-100 (1%), SGB exhibited below 10% hemolysis activity at 100 µg/mL, showing a relatively low hemolysis activity (Figure 5b).

## 3. Discussion

As more antibiotics are rendered ineffective by antibiotic-resistant bacteria, the focus must shift to alternative treatments for infections. MRSA is one of the major causes of antibiotic-resistant infections. According to the results of our previous studies, SGB is considered to be a promising natural antimicrobial agent against MRSA, and it was confirmed to reverse the antibacterial activity of β-lactam antibiotics by downregulating MRSA resistance-related genes [20]. In this study, we evaluated the synergistic effect of SGB with six conventional antibiotics, including non-β-lactam antibiotics, to learn more about its therapeutic utility. The results of the checkerboard assay indicated that SGB showed synergy or partial synergy with all six conventional antibiotics, and none of the tested combinations showed antagonism. Notably, SGB showed synergy with linezolid and vancomycin, the guideline antibiotics for MRSA treatment. This is an encouraging sign that the combination of SGB and antibiotics may be valuable in reducing the use of antibiotics. The synergy was further confirmed in a time-to-kill assay, in which the combined treatment group significantly inhibited bacterial growth after 4 h and nearly eradicated bacteria within 24 h. This result reveals that the combination therapy may have higher and faster bactericidal activity. Metastatic spread of MRSA infection is generally considered to be related to the duration of bacteremia [21], while the combination therapy of SGB and antibiotics may shorten the duration of treatment.

The clinical failure of several conventional antibiotics such as vancomycin in the treatment of persistent pulmonary MRSA by systemic or local administration can be rationalized by incapability to efficiently penetrate biofilms [22]. Interestingly, the combination of vancomycin and SGB effectively inhibited the growth of *S. aureus* strain, with a 2–8-fold reduction in MIC. Therefore, we speculated that SGB may have biofilm-disrupting properties that contribute to the potent activity of the subsequent release of antibacterial agents against protected colonies thriving in the biofilm state [22]. Subsequently, the results of crystal violet biofilm inhibition assays verified our conjecture that SGB inhibited biofilm formation in a dose-dependent manner at subinhibitory concentrations. Moreover, we assumed that SGB inhibits biofilm formation by downregulating the expression of biofilm-related genes, as evidenced by the RT-qPCR results. This finding is also consistent with previous reports that saponins can disrupt the biofilm [23], and they indicate that SGB is a promising lead compound for targeted biofilm therapy.

Furthermore, the limitation of conventional antibiotics for MRSA treatment has also been attributed to their low penetration into cell membranes [24]. Notably, saponin is a natural nonionic detergent with cell membrane permeability, and its hydrophobic structure directly contacts the phospholipid bilayer of microbial cell membranes, leading to enhanced ion permeability and leakage of important intracellular components [25]. We speculated that another mechanism for the synergy between SGB and conventional antibiotics was related to SGB improving the antibacterial efficiency of conventional antibiotics by destroying the cell membrane of MRSA. The findings of the crystal violet absorption assay and TEM gave evidence in support.

We performed a preliminary drug resistance assay, where the MIC value of SGB was not increased significantly after 10 consecutive passages, consistent with the report that the plant-derived compound has little drug resistance (Appendix A, Figure A1). Moreover, the cytotoxicity hemolysis activity was only observed at concentrations far exceeding those necessary for a reversal of conventional antibiotic resistance in MRSA. Therefore, these advantages embody the potential of SGB as an antibiotic adjuvant, which is of great significance for increasing the industrial and medical applications of SGB.

The current study, however, was subject to several limitations. First, the current study was limited to determining the contribution of in vitro experiments. Even if the results of this study yielded a statistically significant synergy, it is insufficient to give clinical recommendations. Future work will be required to investigate the results of in vivo experiments. Second, the addition of a therapeutic agent may raise the likelihood of side-effects. Hence, future investigations on the combination treatment will require close monitoring of renal impairment. Third, the strains used in this study were a standard strain and two clinical strains from the same hospital. Thus, the sample size was small and had regional limitations. Therefore, more diverse clinical isolates should be included in future studies to better elucidate the feasibility of the clinical application of SGB in synergistic therapy with conventional antibiotics.

In summary, the small scale of this study limited the ability to draw strong conclusions; however, it serves as a pilot trial, provides a valuable reference for the development of therapeutic methods for preventing the development of drug resistance, and lays the foundation for the development of targeted biofilm drugs.

## 4. Materials and Methods

### 4.1. Reagents

SGB (Figure 6) was isolated from *Sanguisorba officinalis* L., and identified by spectral and physicochemical methods; the purity of HPLC analysis was more than 98% [20]. Skim milk, Mueller–Hinton agar (MHA), and Mueller–Hinton broth (MHB) were purchased from Difco Laboratories (Baltimore, MD, USA). Crystal violet, linezolid, gentamicin, vancomycin, amikacin, amoxicillin, ceftazidime, and dimethyl sulfoxide (DMSO) were obtained from Sigma-Aldrich Co. (St. Louis, MO, USA). The E.Z.N.A. Bacterial RNA Kit was obtained from Omega Bio-Tek (Norcross, GA, USA). The sequences of primers used in qRT-PCR was purchased from Bioneer (Daejeon, Korea).

### 4.2. Bacterial Strains

In this study, *S. aureus* ATCC 33591 (American Type Culture Collection, Manassas, VA, USA) was used as a reference strain. MRSA (DPS-1 and DPS-3) were isolated from patients at the Hospital of Wonkwang University and used as clinical isolates. *S. aureus* was cultured in MHA or MHB at 37 °C.

### 4.3. Checkerboard Assay

A checkerboard assay was used to assess the MIC values of SGB in combination with conventional antibiotics (linezolid, gentamicin, vancomycin, amikacin, amoxicillin, and ceftazidime) according to the Clinical and Laboratory Standards Institute (CLSI) standards [25]. *S. aureus* strains (ATCC 33591, DPS-1, DPS-3) were grown on MHA for 24 h at 37 °C. In Mueller–Hinton broth, serial dilutions of SGB with antibiotics were combined (MHB). MRSA inocula were adjusted in MHB to the 0.5 McFarland standard. The final inoculum had a bacterial concentration of 1.5 × 10^5^ CFU/well. After a 24 h incubation period at 37 °C, each MIC value was determined and defined as the lowest concentration that inhibited the growth of the *S. aureus*. The fractional inhibitory concentration index (FICI) was used to determine the interaction between SGB and conventional antibiotics as follows:∑FIC: FICA + FICB = MICA + B/MICA alone + MICB + A/MICB alone.

The combination was considered as synergy for FICI ≤ 0.5, partial synergy for 0.5 < FICI ≤ 0.75, an additive effect for 0.75 < FICI ≤ 1, indifference for 1 < FICI ≤ 4, and antagonism for FICI > 4. In addition, the fold reduction in the MIC of antibiotics against MRSA alone or in combination with SGB was calculated, shown in Table 1 as Fold. All tests were performed three times.

### 4.4. Time-to-Kill Assays

A time-to-kill assay was performed to further determine the synergistic antimicrobial effect. The method was carried out according to the recommendations of CLSI. Bacterial cultures were diluted with MHB to 1.5 × 10^5^ CFU/mL and incubated at 37 °C for 24 h. Antimicrobial agent concentrations were set at subinhibitory concentrations (1/2 MIC) in both single and combination treatments. SGB was combined with six antibiotics against MRSA (reference strain ATCC 33591 and clinical isolate DPS-1) for the combined treatment groups. Bacterial growth curves were observed at five different timepoints (0, 4, 8, 16, and 24 h).

### 4.5. Crystal Violet Biofilm Assay

The SGB inhibition of the biofilm formation of *S. aureus* was performed in a previous study [15], with two *S. aureus* strains DPS-1 and ATCC 33591. Briefly, 100 μL of overnight culture (0.5 MacFarland bacterial culture) was added to each well of 96-well microtiter plates and treated with subinhibitory concentrations of SGB. The planktonic cells were removed after 24 h at 37 °C and washed three times with PBS, and each well of the 96-well microtiter plates was stained with 1% (*w*/*v*) crystal violet for 10 min at room temperature before being rewashed three times. The stained biofilms were solubilized in 100 μL of absolute ethanol, and the optical density (OD) values at 570 nm were determined. Using the formula below, the percentage biofilm inhibition was calculated.

Percentage inhibition = 100 − ((OD 570 nm of the treatment wells)/(OD 570 nm of the control wells) × 100)).

### 4.6. Quantitative RT-PCR (qRT-PCR)

A qRT-PCR was performed using a previously published procedure [26]. MRSA (ATCC 33591 and DPS-1) was grown overnight in MHB and then treated for 1 h with subinhibitory concentrations of SGB. As a control, a sample without SGB was used. Total RNA was extracted from *S. aureus* using the E.Z.N.A.^®^ bacterial RNA kit according to the manufacturer’s instructions (Omega Bio-tek, Norcross, GA, USA). A spectrophotometer was used to determine the RNA concentration by measuring the absorbance ratio at 260 nm (BioTek, Winooski, VT, USA). The complementary DNA was then synthesized using a QuantiTect reverse transcription kit (Qiagen, Dusseldorf, Germany) according to the manufacturer’s instructions. Thus, 2 μL of sample cDNA, 1 μL of each primer (10 L/mL), 6 μL of deionized water, and 10 μL of 2 SYBR Green PCR master mix (Life Technologies LTD, Warrington, UK) were used in a total volume of 20 μL. The delta–delta cycle threshold method was used to calculate the expression level of the target gene relative to the endogenous reference gene 16 rRNA using StepOne software v2.3 from Applied Biosystems (Waltham, MA, USA). Primer sequences used in this study were as follows: *16S* (5′–3′) F: ACTCCTACGGGAGGCAGCAG, R: ATTACCGCGGCTGCTGG; *hld* (5′–3′) F: ATTTGTTCACTGTGTCGATAATCC, R: GGAGTGATTTCAATGGCACAAG.

### 4.7. Crystal Violet Absorption Assay

A crystal violet assay was used to detect the alteration in membrane permeability [27]. Suspensions of the MRSA (ATCC 33591) were prepared in MHB. Centrifugation at 4500× *g* for 5 min at 4 °C was used to harvest cells. The cells were washed twice with PBS and resuspended. Vancomycin and SGB were added to the cell suspension at 2 MIC and 4 MIC concentrations and incubated for 30 min at 37 °C. Samples without treatment were similarly prepared with SGB as a control. Cells were harvested at 9300× *g* for 5 min. The cells were then resuspended in PBS with 10 µL/mL crystal violet. After that, the cell suspension was incubated for 10 min at 37 °C. After centrifuging the sample at 13,400× *g* for 15 min, the OD_590_ of the supernatant was determined using a spectrophotometer. The OD value of the crystal violet solution, which was originally utilized in the experiment, was determined to be 1. The crystal violet absorption in all samples was estimated using the following formula:

Crystal violet absorption = (OD value of the sample/OD value of the crystal violet solution) × 100.

### 4.8. Transmission Electron Microscopy (TEM)

MRSA exponential phase cultures were obtained by overnight dilution in MHB and continued growth at 37 °C until the cultures reached the mid-logarithmic phase of growth. The MHB-grown exponential-phase MRSA ATCC 33691 was treated with 1/2 MIC and the MIC of SGB for 4 h. Following treatment, 2 mL of the culture was centrifuged at 10,000× *g* for 10 min to collect. After removing the supernatant, pellets were fixed by immersion in a modified Karnovsky fixative solution containing 2% paraformaldehyde and 2% glutaraldehyde in a sodium cacodylate buffer solution of 0.05 M (pH 7.2) [28]. A 4K slow-scan charge-coupled device camera (Ultrascan 4000 SP; Gatan, Pleasanton, CA, USA) linked to an electron microscope was used to record transmitted electron signals.

### 4.9. Cytotoxicity Assay

In vitro cytotoxicity was assessed in RAW 264.7 cells using the CellTiter 96AQueous One Solution Reagent (Promega) following the procedure in [29]. RAW 264.7 cells were cultured at 37 °C in an atmosphere of 5% CO_2_ and seeded at 5 × 10^4^ cells per well in a total volume of 100 μL in 96-well plates. After 24 h, the medium was replaced with fresh medium containing a series of concentrations of SGB (2 MIC, 4 MIC, and 8 MIC). Here, 0.5% DMSO was used as the control. Each culture well was optimized using a microplate reader (Titertek Multiskan, Flow Laboratories, North Ryde, Australia) at 490 nm after the medium was replaced with an MTS solution. The formula for calculating cell viability was as follows:

Cell viability (%) = (OD_490_ value of CTT treated cells/OD_490_ value of untreated cells) × 100.

### 4.10. In Vitro Hemolysis Assay

A hemolysis assay was performed to assess the toxic hemolysis of the drug according to the product protocol. Rabbit blood was washed with PBS until the supernatant was clear. A series of SGB solutions were incubated with a 2% blood solution at 37 °C for 30 min. Triton X-100 (1%) was used as a positive control, and 0.5% DMSO was used as a negative control. After incubation, the mixture was centrifuged at 2500× *g* for 6 min. Afterward, 100 uL of supernatant from each sample was placed in a 96-well plate, and the absorbance at 541 nm was measured.

### 4.11. Statistical Analysis

Analyses were performed in triplicate, and the results were reported as the mean standard deviation. An independent Scheffe’s *t*-test was used to statistically assess the acquired data (SPSS software version 22.0; IBM SPSS, Armonk, NY, USA). A *p*-value of less than 0.05 was considered statistically significant.

## 5. Conclusions

The current study evaluated the synergistic effect and synergistic mechanism of sanguisorbigenin (SGB) with six conventional antibiotics. We investigated the inhibitory effect of SGB on biofilm formation and the effect on cell membrane permeability.

## Figures and Tables

**Figure 1 ijms-23-04232-f001:**
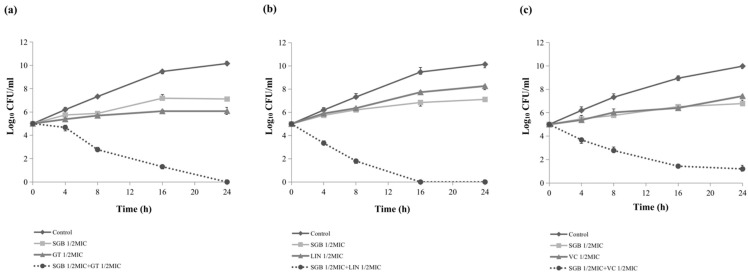
Time-to-kill curves showing synergistic interaction between SGB and conventional antibiotics against *S. aureus* ATCC 33591 (**a**–**c**) and DPS-1 (**d**–**f**). SGB, sanguisorbigenin; GT, gentamicin; LIN, linezolid; VC, vancomycin, CEF, ceftazidime, AMO, Amoxicillin, AMK, amikacin. Data are the mean ± standard deviation of three independent experiments.

**Figure 2 ijms-23-04232-f002:**
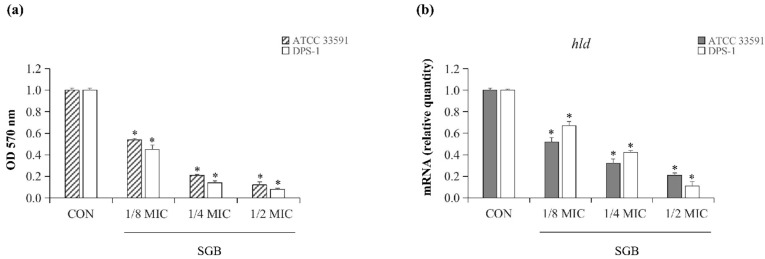
(**a**) Inhibition of SGB at subinhibitory concentrations on MRSA (ATCC 33591, DPS-1) biofilm. (**b**) The expression of *hld* in MRSA (ATCC 33591, DPS-1) cultures exposed to SGB at subinhibitory concentrations. The data are presented as the mean ± standard deviation of three independent experiments. * *p* < 0.05.

**Figure 3 ijms-23-04232-f003:**
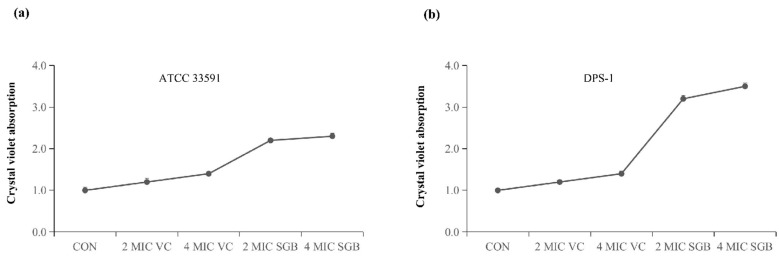
Crystal violet absorption of SGB and vancomycin-treated *S. aureus*; vancomycin was used as a negative control. (**a**) The crystal violet absorption of *S. aureus* ATCC 33591 treated with 2 MIC and 4 MIC of SGB or vancomycin. (**b**) The crystal violet absorption of *S. aureus* DPS-1 treated with 2 MIC and 4 MIC of SGB or vancomycin. SGB, sanguisorbigenin; VC, vancomycin. The data are presented as the mean ± standard deviation of three independent experiments.

**Figure 4 ijms-23-04232-f004:**
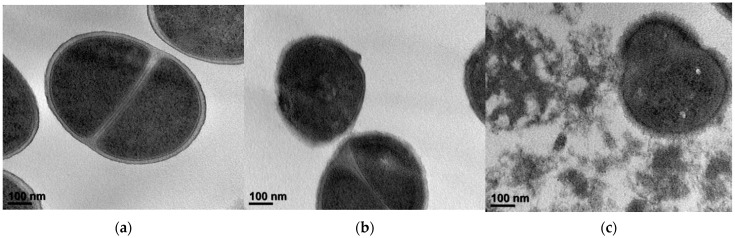
TEM images of MRSA (ATCC 33591) after treatment with SGB. (**a**) Untreated control MRSA. (**b**) MRSA treated with 1/2 MIC of SGB (6.3 µg/mL). (**c**) MRSA treated with the MIC of SGB (12.5 µg/mL).

**Figure 5 ijms-23-04232-f005:**
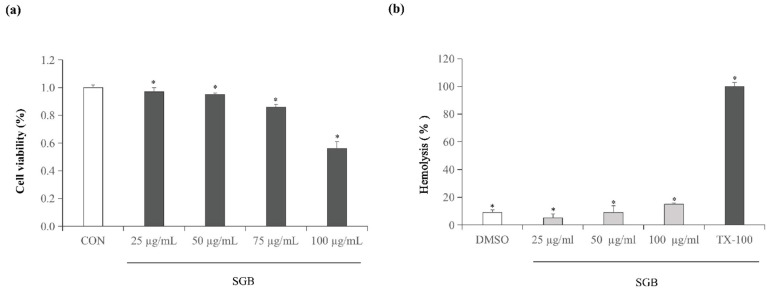
(**a**) Effects of SGB treatment on cytotoxicity in RAW 264.7 cells. Cell viability was evaluated using an MTS assay after 24 h treatment with SGB. (**b**) Hemolysis of SGB on rabbit blood cells. Triton X-100 (TX-100) was used as a positive control. Dimethyl sulfoxide (DMSO) was used as a negative control. The data are presented as the mean ± standard deviation of three independent experiments. * *p* < 0.05.

**Figure 6 ijms-23-04232-f006:**
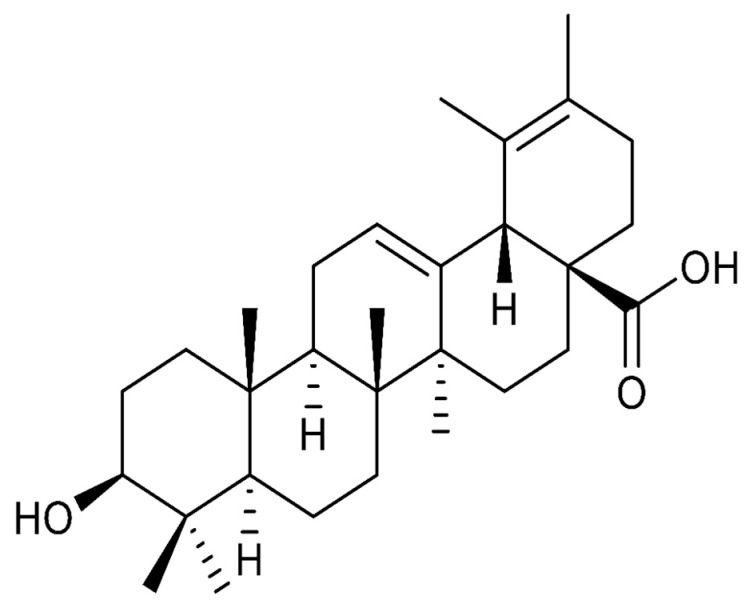
Chemical structure of SGB.

**Table 1 ijms-23-04232-t001:** Synergistic effect of SGB combined with conventional antibiotics and the MIC of antibiotics used alone or combination.

Antibiotics	ATCC 33591			DPS-1			DPS-3		
	MIC (μg/mL) (Alone/ Combination)	Fold	FICI	MIC (μg/mL) (Alone/ Combination)	Fold	FICI	MIC (μg/mL) (Alone/ Combination)	Fold	FICI
Linezolid	1.9/0.9	2	1	500/62.5	8	0.25	250/62.5	4	0.38
Gentamicin	3.9/0.9	4	0.38	125/15.6	8	0.25	250/15.6	16	0.19
Vancomycin	1.9/0.9	2	1	250/62.5	4	0.5	500/62.5	8	0.25
Amikacin	62.5/31.3	2	1	125/31.3	4	0.5	125/31.3	4	0.75
Amoxicillin	125/62.5	2	1	125/62.5	2	1	250/62.5	4	0.5
Ceftazidime	31.3/7.8	4	0.5	31.3/3.9	8	0.63	62.5/7.8	8	0.25

MIC, minimal inhibitory concentration; MIC values were shown for antibiotics alone or in combination with SGB; Fold, fold reduction in antibiotic MIC with combination therapy; FICI, fractional inhibitory concentration index.

## Data Availability

Personal information is included; thus, data are available for research only.

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
