# Peer review of "Combination of Sanguisorbigenin and Conventional Antibiotic Therapy for Methicillin-Resistant Staphylococcus aureus: Inhibition of Biofilm Formation and Alteration of Cell Membrane Permeability"

_ijms, 2022, doi:10.3390/ijms23084232_

Round 1
Reviewer 1 Report
Well written paper with great method explanation.
My main comment is regarding figure 1:
What is your detection limit and dynamic range for CFU counts?
Figures show log of 0 which is infinite. Authors need to start their vertical axis from where their detection limit is. For example if your colony counter is able to count 100 CFUs then your detection limit is 2. Please correct all the corresponding graphs.
Author Response
"Please see the attachment"

Reviewer 2 Report
The paper uses solid methology and is overall of interest, however, I am not entirely convinced that the number of methods used, the only once compound (sanguisorbigenin) tested and the limited number of isolates enrolled in the study allow for far-reaching conclusions that would warrant publication in this high-impact and high-quality journal. Overall, the novelty and significance of the results needs to be further highlighted/clarified, otherwise the MS should not be considered for publication.
General:
biological names (bacteria, plants) should be in italics, please follow international rules for taxonomy and nomenclature
latin terms (in vitro, in vivo) should be in italics
- Introduction
L33-34: please elaborate on the epidemiology of MRSA in the context of human infections
L39-L41: this sentence is very akward and repetitive, please rephrase it
L42-44: the authors should elaborate more on the global significance of AMR using current literature sources
L45-55: the authors should elaborate more on the composition of bacterial biofilms and their significance, as the present version of this part is quite basic. please discuss the correlation between biofilm-formation and MDR in staphylococci using current literature sources
- Results
the data presented in Table 1. is akward and hard to read
Figures 1.-3. are of poor quality and visibility, they need to be improved
panes from Fig. 4 need to be enlarged for better visibility
Discussion:
L163-165: this is a strong statement, which has – in my opition – not been entirely proven by the present results
L192-194: why didnt you inlcude these results in the present ms?
Methods:
for the assessment of combination, more current indices (e.g., CI, combination index) have been proposed, which better reflect biological interactions between compounds
Author Response
"Please see attachment"

Round 2
Reviewer 2 Report
The authors have addressed most of my concerns regarding the initial version of the manuscript.
Please consider including a very recent and comprehensive review on AMR in the introduction at L45-54:
https://www.sciencedirect.com/science/article/pii/S1876034121003403